# Mid-upper arm circumference as a simple tool for identifying central obesity and insulin resistance in type 2 diabetes

Yanhua Zhu[1☯], Qiongyan Lin[2☯], Yao Zhang[1], Hongrong Deng[1], Xiling Hu[1], Xubin Yang[1]*, Bin Yao[1]*

**1** Department of Endocrinology, the Third Affiliated Hospital of Sun Yat-sen University, Guangzhou, China, **2** Department of Endocrinology, Jieyang People's Hospital (Jieyang Affiliated Hospital, Sun Yat-sen University), Jieyang, Guangdong, China

☯ These authors contributed equally to this work.
* binyao1910@126.com (BY); shansheep@126.com (XY)

## Abstract

### Background

Our research aimed to explore the correlation between mid-upper arm circumference (MUAC) and central obesity and insulin resistance (IR) in Chinese subjects with type 2 diabetes.

### Materials

A total of 103 participants (60 men) were recruited in our study. MUAC was measured around the mid-arm between the shoulder and elbow. Waist circumference (WC) was obtained as central obesity parameter, and the IR parameter of Homeostasis Model Assessment-Insulin Resistance (HOMA-IR) was calculated. The subjects were divided into three groups according to the tertiles cut-points of MUAC level.

### Results

Body mass index (BMI), WC, the percentages of central obesity and HOMA-IR were significantly higher in the groups with higher MUAC than those in the group with lower MUAC (all $P < 0.05$). Pearson analysis showed that MUAC was correlated with BMI, WC, waist-to-hip ratio (WHR), logHOMA-IR, low density lipoprotein cholesterol (LDL-C), uric acid (UA) and high density lipoprotein cholesterol (HDL-C) in all subjects. Multivariate linear regression analysis revealed that MUAC was independently associated with logHOMA-IR ($\beta = 0.036$, $P<0.001$) after adjusting for age, gender, WHR, UA, TG, LDL-C and HDL-C. Binary logistic regression analysis revealed that MUAC was an independent predictor of central obesity (OR: 2.129, 95%CI: 1.311–3.457, $P = 0.002$). Furthermore, MUAC≥30.9cm for male and ≥30.0cm for female were the optimal cutoff values for identifying central obesity.

**Data Availability Statement:** All relevant data are within the paper and its Supporting Information files.

**Funding:** This work was supported by Science and technology planning project of Guangzhou Tian He district(2018YT016); Natural Science Foundation of Guangdong Province [2018A030313915]; Medical Scientific Research Foundation of Guangdong Province of China [A2018286]; the National Key Research and Development Program [2017YFC1309602]; and Science and Technology Program YueXiu District (2018-WS-005).

**Competing interests:** The authors have declared that no competing interests exist.

## Conclusions

Our study indicated that among Chinese subjects with type 2 diabetes, MUAC is a simple and effective tool for the determination of central obesity and IR. Additionally, the larger MUAC is proved to be more associated with metabolic risk factors of higher UA and LDL-C and lowewer HDL-C.

## Introduction

Obesity is an international issue related to many serious diseases like diabetes and cardiovascular diseases that impose a huge burden on both individual and public health [1–3]. Based on international reference standards, the body mass index (BMI) is the most common measurement for the determination of obesity both in clinical practice and research [4,5]. However, more and more studies confirmed that central obesity, also known as visceral obesity, offers more predictive power for type 2 diabetes, cardiovascular risk and metabolic dysfunctions than whole-body adiposity [6–8]. Thus, for the failure to evaluate body fat distribution, BMI is replaced by some other anthropometric parameters to predict central obesity and insulin resistance, such as waist circumference (WC) and waist to hip ratio (WHR) [9–12].

Among various methods, WC is used as the most common anthropometric index of abdominal visceral fat accumulation and insulin resistance (IR), which were the indicators of cardiovascular risks in both men and women [3,13]. However, there remain a number of limitations of WC, such as the absence of a standard approach of measurement; the volatility of measuring results from the influence of dining and diverse health conditions [10]. In addition, while it is well documented that adipose tissue, especially volume of visceral adipose tissue was strongly correlated with cardiovascular diseases, insulin resistance and diabetes mellitus [14], WC alone to predict central obesity seems to be not enough since its failure to distinguish whether it is caused by volume of visceral or abdominal subcutaneous adipose tissue. Therefore, some researches were conducted to explore novel indexes to be more accurate and practical [10,15–17]. Recently, arm-fat percentage or mid-upper arm circumference (MUAC) were suggested as novel predictors of central obesity and IR in population with normal weight, overweight or obesity [18–20]. However, there is little data to evaluate the role of MUAC in detecting IR and central obesity in diabetic patients. A large amount of studies had proved that the IR and central obesity in patients with diabetes were quite different from other populations [21,22]. Type 2 diabetes is mainly characterized by insulin resistance. Most recently, in Groop L's and Ji L's suggestions of novel diabetes classification, diabetes was stratified into five types (in Groop L's study) or four types (in Ji L's study) [23,24]. In their studies, diabetes complications substantially increased in patients with severe insulin-resistant diabetes, which reinforced the importance of IR and central obesity in diabetic patients. As a result, the association of MUAC and IR and central obesity in diabetes might be quite different from other populations as well. Therefore, precisely identifying the central obesity and IR in closely related to type 2 diabetes is of great importance. As the evaluation of MUAC can be easily obtained in clinical practice, our study aimed to evaluate whether the MUAC can be served as an indicator of central obesity and IR in type 2 diabetes. Besides, we further investigate whether MUAC is superior to other anthropometric parameters in measuring central obesity and IR in subjects with type 2 diabetes.

## Materials and methods

### Study population

From April 2015 to May 2017, 103 subjects were recruited in our study. The patients were selected from inpatient clinics who met the following criteria: aged above 18 years, diagnosed with type 2 diabetes (according to the WHO diabetes criteria) [25], without insulin treatment, Sulphonylurea, sodium-dependent glucose transporters-2 inhibitors (SGLT-2i) or glucagon-like peptide-1 (GLP-1) analog for at least 2 weeks (hypoglycemic agents in S1 Table), not having severe disease and be free of any acute infection during 2 weeks before the inclusion. The medications of hypertension, hyperlipidemia and hyperuricemia were displayed in the S2–S4 Tables, respectively. The protocol was approved by the ethics committee of the Third Affiliated Hospital of Sun Yat-sen University. All subjects provided written informed consent before screening.

### Anthropometric measures

Body height and weight were measured by the researchers and BMI was calculated as body weight(kg) divided by the square of the height(m). Hip circumference (HC) was the horizontal length between the most prominent parts of the buttocks, waist circumference (WC) was measured at the mid-position between the iliac crest and the last rib [26], and the waist-to-hip ratio (WHR) was calculated. Mid-upper arm circumference (MUAC) was measured at the mid-arm between the shoulder and elbow [18].

### Blood biochemical assays

Venous blood samples were collected from participants to determine metabolic markers, including fasting blood-glucose, total cholesterol (TC), low density lipoprotein cholesterol (LDL-C), high density lipoprotein cholesterol (HDL-C), triglyceride (TG), and uric acid (UA), using an automated enzymatic method (Hitachi, Japan, 7600–020 autonomic analyzer). High-pressure liquid chromatography (BIO-RAD, USA, D-10 analyzer) was used to measure the HbA1c level. Plasma insulin was evaluated by competitive radioimmunoassay (Centaur XP immunoassay system, Siemens Healthcare Diagnostics, New York, NY). The Homeostasis Model Assessment-Insulin Resistance (HOMA-IR) index, calculated by the formula: fasting plasma insulin (mU/L) x fasting plasma glucose (mmol/L)/22.5 [27], was used to assess Insulin Resistance (IR).

For man, waist circumference $\geq$ 85cm was used to define central obesity and for women as waist circumference $\geq$ 80cm [28].

### Statistical analysis

The SPSS program for Windows (version 22.0) was utilized to process the analysis. Continuous variables of clinical characteristics were presented as mean±S.D. Differences between groups were evaluated by t-test, chi-square test or Wilcoxon rank-sum test. Data of non-normal distributions were logarithmically transformed before statistical analysis. Pearson or Spearman correlations were conducted to analyze the associations between two variables. Variables that were significantly related to the objective variable were tested for independence using multivariate linear regression analysis. The risk factors of central obesity were evaluated by logistic regression. The accuracy of anthropometric parameters to predict central obesity was evaluated by using receiver operator characteristic (ROC) curve analysis. The specificity and sensitivity of MUAC were calculated for each cut-off point in the sample. P value below 0.05 was examined statistically significant.

**Table 1. General characteristics of study subjects.**

| Parameters | All(N = 103) | Men(N = 60) | Women(N = 43) | P(men vs women) |
|---|---|---|---|---|
| Age (years) | 51.4±13.6 | 48.9±14.2 | 54.9±12.2 | 0.03 |
| Diabetes duration (years) | 8.79±3.13 | 9.82±3.21 | 8.13±2.11 | 0.78 |
| BP (mmHg) | 135/83 | 145/92 | 134/80 | 0.23 |
| FBG (mmol/L) | 8.70±3.34 | 8.62±2.79 | 8.83±4.02 | 0.75 |
| BMI (kg/m$^2$) | 25.32±4.05 | 26.27±4.58 | 24.00±2.73 | 0.005 |
| WC (cm) | 91.51±9.62 | 93.78±10.67 | 88.35±6.88 | 0.004 |
| WHR | 0.94±0.06 | 0.95±0.05 | 0.93±0.06 | 0.13 |
| MUAC (cm) | 31.68±3.35 | 32.82±3.38 | 30.11±2.62 | <0.001 |
| TG (mmol/L) | 2.15±2.55 | 2.23±2.13 | 2.04±3.06 | 0.71 |
| TC (mmol/L) | 5.07±1.61 | 4.77±1.28 | 5.47±1.91 | 0.03 |
| HDL-C (mmol/L) | 1.06±0.30 | 0.96±0.23 | 1.21±0.33 | <0.001 |
| LDL-C (mmol/L) | 3.17±1.06 | 3.18±0.99 | 3.17±1.15 | 0.95 |
| UA (μmol/L) | 354.17±99.49 | 368.20±100.16 | 334.60±96.29 | 0.09 |
| HbA1C (%) | 9.04±2.51 | 9.16±2.30 | 8.86±2.79 | 0.56 |
| LogHOMA-IR | 0.61±0.23 | 0.64±0.23 | 0.57±0.22 | 0.13 |

BMI, Body Mass Index; MUAC, Mid-upper Arm Circumference; WC, Waist Circumstance; WHR, Waist-to-hip Ratio; UA, Uric Acid; TG, Triglyceride; TC, Total Cholesterol; HDL-C, High-Density Lipoprotein Cholesterol; LDL-C, Low-Density Lipoprotein Cholesterol; HbA1C, Hemoglobin A1c; HOMA-IR, Homeostatic Model Assessment of Insulin Resistance.

## Results

### Baseline characteristics

The study sample consisted of 103 individuals (mean age: 51.4±13.6 years) with 60 male (58%). The clinical characteristics of the subjects are shown in Table 1. The values of BMI (male, 26.27±4.58kg/m2, female, 24.00±2.73kg/m2), WC (male, 93.78±10.67cm, female, 88.35 ±6.88cm) and MUAC (male, 32.82±3.38cm, female, 30.11±2.62cm) were significantly higher while TC (male, 4.77±1.28mmol/L, female, 5.47±1.91mmol/L) and HDL-C (male, 0.96 ±0.23mmol/L, female, 1.21±0.33mmol/L) were significantly lower in male than those in female.

In addition, the subjects were categorized based on tertiles cut-points of arm circumference level (tertile 1, MUAC < 29.50cm; tertile 2, 29.50cm≤MUAC≤32.60cm; tertile 3, MUAC > 32.60cm). As shown in Table 2, the value of BMI, WC and LogHOMA-IR of patients were significantly higher in the groups with higher MUAC. Moreover, the percentages of central obesity, identified by the measurement of waist circumference, were 55.8%, 85.7% and 97.1% respectively in the three groups (*P*<0.01).

### Correlation of mid-upper arm circumference with central obesity parameters, metabolic variables and insulin resistance

Correlation between the MUAC, central obesity markers and metabolic markers among all subjects were analyzed in our study. The MUAC positively correlated with BMI (r = 0.88, *P* < 0.001) and the central obesity markers, WC (r = 0.74, *P* < 0.001) and WHR (r = 0.30, *P* = 0.002). Among the traditional metabolic risk factors, MUAC showed a positive correlation with UA(r = 0.32, *P*<0.001) and LDL-C (r = 0.40, *P*<0.001) and negatively correlated with HDL-C (r = -0.32, *P* = 0.001) in all patients (Fig 1). Furthermore, the relationships shown in Table 3 were observed differently by gender. In both genders, MUAC correlated positively

**Table 2. Characteristics of subjects according to mid-upper arm circumference tertiles.**

| | Tertile 1 (n = 34) | Tertile 2 (n = 35) | Tertile 3 (n = 34) | P value | | |
| --- | --- | --- | --- | --- | --- | --- |
| | | | | 1 vs 2 | 1 vs 3 | 2 vs 3 |
| Age(years) | 55.2±11.7 | 52.1±12.5 | 46.9±12.5 | 0.33 | 0.01 | 0.11 |
| BMI(kg/m$^2$) | 21.97±1.07 | 24.51±1.27 | 29.52±4.21 | <0.01 | <0.01 | <0.01 |
| MUAC (cm) | 28.36±0.90 | 31.26±0.87 | 35.47±2.62 | - | - | - |
| WC (cm) | 84.59±7.07 | 89.94±5.32 | 100.04±8.93 | <0.01 | <0.01 | <0.01 |
| WHR | 0.92±0.07 | 0.95±0.04 | 0.97±0.06 | 0.09 | <0.01 | 0.13 |
| HbA1C (%) | 9.29±3.01 | 8.98±2.26 | 8.84±2.15 | 0.62 | 0.60 | 0.83 |
| FBG (mmol/L) | 9.40±4.38 | 8.26±270 | 8.45±2.64 | 0.16 | 0.24 | 0.83 |
| BP (mmHg) | 135/94 | 142/98 | 138/92 | 0.24 | 0.35 | 0.48 |
| TC (mmol/l) | 4.96±1.32 | 5.02±1.65 | 5.22±1.85 | 0.90 | 0.61 | 0.53 |
| TG (mmol/l) | 1.36±0.71 | 2.17±2.01 | 2.93±3.76 | 0.18 | 0.01 | 0.21 |
| HDL-C (mmol/l) | 1.24±0.34 | 1.01±0.21 | 0.95±0.26 | <0.01 | <0.01 | 0.34 |
| LDL-C (mmol/l) | 2.85±0.85 | 2.93±1.21 | 3.74±0.85 | 0.36 | <0.01 | 0.02 |
| UA(umol/L) | 312.06±74.04 | 358.51±106.79 | 391.82±100.27 | 0.05 | <0.01 | 0.15 |
| LogHOMA-IR | 0.46±0.21 | 0.63±0.17 | 0.74±0.21 | <0.01 | <0.01 | 0.02 |

For list of abbreviations, see Table 1.

P value <0.05 were considered significant.

with BMI, WC and LDL-C, while the positive correlation with WHR, TG, UA and negative correlation with HDL-C were only shown in men. The association between MUAC and glycemic parameters was not found in our research.

Our results showed that MUAC (male, r = 0.52, *P*<0.01, female, r = 0.55, *P*<0.01), WC (male, r = 0.44, *P*<0.01, female, r = 0.35, *P* = 0.02) and BMI (male, r = 0.46, *P*<0.01, female, r = 0.51, *P*<0.01) were positively related to LogHOMA-IR in both genders. However, we found that WHR (male, r = 0.33, *P* = 0.01, female, r = 0.74, *P* = 0.64) significantly positively correlated with LogHOMA-IR only in male. Additionally, MUAC was more strongly correlated with LogHOMA-IR than WC, BMI and WHR in both genders (Table 4).

Linear regression analysis was conducted to identify the predictive effect of MUAC on the insulin resistance. After adjusting for confounding clinical parameters, including age, gender, WHR, UA, TG, LDL-C and HDL-C, the value of MUAC remained independently associated with LogHOMA-IR (β = 0.036, *P*<0.001) (Table 5).

The results of logistic regression analysis presented in Table 6 showed that only MUAC (OR, 2.129; 95% CI, 1.311–3.457; *P* = 0.002) and LDL-C (OR, 3.023; 95% CI, 1.090–8.383; *P* = 0.033) were associated with increased odds of central obesity after adjusting for age, gender, TG, AC,UA and use of statin.

## Optimal cut-off points of mid-upper arm circumference for central obesity

The proportion of central obesity was 81.7% in men and 83.7% in women. The ROC curves are presented in Fig 2, and the area under the curve (AUC) for MUAC as a predictor of central obesity was 0.922 for male and 0.788 for female. The best MUAC cutoff point for defining central obesity was 30.9cm for male (Youden index = 0.746, Sensitivity: 83.7%; Specificity: 90.9%) and 30.0cm for female (Youden index = 0.528, Sensitivity: 52.8%; Specificity: 99.9%).

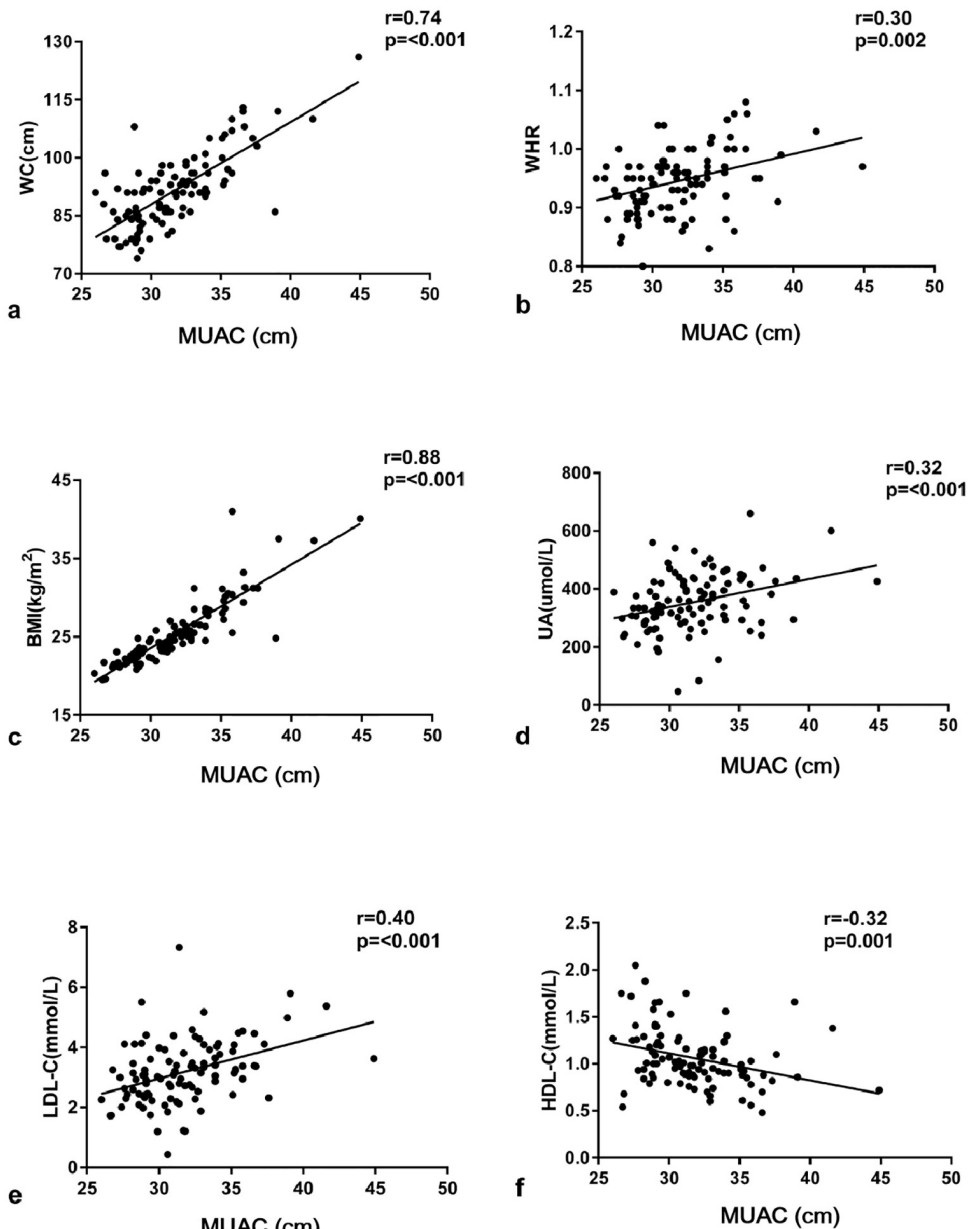

**Fig 1. Relationship between MUAC (cm) and other anthropometric measurements in all subjects.** Correlation assessed by Pearson analysis. MUAC, Mid-upper Arm Circumference; WC, Waist Circumstance; WHR, Waist-to-hip Ratio; BMI, Body Mass Index; UA, Uric Acid; LDL-C, Low-Density Lipoprotein Cholesterol; HDL-C, High-Density Lipoprotein Cholesterol.

## Discussion

Previous studies [6,7,29–31] in different subjects demonstrated that abdominal distribution of body fat was correlated with IR, type 2 diabetes, cardiovascular diseases and total mortality risk. And the hyperglycemia was more difficult to manage in individuals with both diabetes and central obesity compared with those with only diabetes [32]. As a result, to found out an appropriate and more easily conducted method to identity central obesity and subsequently

**Table 3. Relationship between MUAC and other measurements by gender.**

| Variables | MUAC (cm) | | | |
|---|---|---|---|---|
| | Men | | Women | |
| | r | P | r | P |
| BMI (kg/m$^2$) | 0.95 | < 0.01 | 0.92 | < 0.01 |
| WC (cm) | 0.82 | < 0.01 | 0.48 | < 0.01 |
| WHR | 0.48 | < 0.01 | 0.10 | 0.50 |
| TG (mmol/L) | 0.30 | 0.02 | 0.20 | 0.20 |
| TC (mmol/L) | 0.19 | 0.13 | 0.07 | 0.65 |
| HDL-C (mmol/L) | -0.33 | 0.01 | -0.23 | 0.14 |
| LDL-C (mmol/L) | 0.57 | < 0.01 | 0.31 | 0.04 |
| UA (μmol/L) | 0.39 | < 0.01 | 0.21 | 0.17 |
| HbA1C (%) | -0.18 | 0.17 | 0.07 | 0.65 |
| FBG (mmol/L) | -0.06 | 0.67 | 0.03 | 0.83 |
| LogHOMA-IR | 0.52 | < 0.01 | 0.55 | < 0.01 |

For list of abbreviations, see Table 1.

P values<0.05 were considered significant.

choose appropriate hypoglycemic agents could lead to better control of hyperglycemia and eventually reduce the mortality in diabetes. In Rerksuppaphol S' study, MUAC was suggested to be a simple and accurate parameter to identify overweight and obesity in Thai school-age children [33]. In an observational, multinational cross-sectional study with 7337 children aged 9–11 years, MUAC was shown to be a suitable method to detect obesity in children [34]. Moreover, Mazıcıoğlu and his group has demonstrated that MUAC could be a useful measurement in screening body fat distribution in children [35]. However, the subjects of those studies were all not patients with diabetes. In our study, we focused on the MUAC especially in type 2 diabetic patients and found that MUAC could serve as a simple and practical tool to better screen and identify central obesity among type 2 diabetic patients. As shown in the results, both the value of WC and the percentage of patients with central obesity were larger in the group with higher MUAC than those in the group with lower MUAC, which indicated the association between MUAC and central obesity in diabetes. In addition, MUAC correlated well with other anthropometric measurements in central obesity patients with diabetes. In both genders, MUAC was significantly associated with WC and BMI, which indicates that MUAC could be used as an effective indicator for both central obesity and overall obesity. Furthermore, in our study, gender-specific cut-off points of the MUAC for central obesity were also established.

**Table 4. Relationship between HOMA-IR and other anthropometric measurements by gender.**

| Variable | LogHOMA-IR | | | |
|---|---|---|---|---|
| | Men | | Women | |
| | r | p | r | p |
| MUAC (cm) | 0.52 | < 0.01 | 0.55 | < 0.01 |
| WC (cm) | 0.44 | < 0.01 | 0.35 | 0.02 |
| WHR | 0.33 | 0.01 | 0.74 | 0.64 |
| BMI (kg/m$^2$) | 0.46 | < 0.01 | 0.51 | < 0.01 |

For list of abbreviations, see Table 1.

P values<0.05 were considered significant.

**Table 5. Linear regression analysis of logHOMA-IR with different clinical characteristics.**

|  |  | β | P | 95% CI | $R^2$ |
|---|---|---|---|---|---|
| MUAC (cm) | Unadjusted model | 0.039 | <0.001 | 0.028,0.051 | 0.319 |
|  | Model 1 | 0.041 | <0.001 | 0.029,0.054 | 0.327 |
|  | Model 2 | 0.041 | <0.001 | 0.027,0.054 | 0.328 |
|  | Model 3 | 0.036 | <0.001 | 0.021,0.050 | 0.392 |

Model 1: adjusted for age and gender.

Model 2: adjusted for age, gender and Waist-to-hip Ratio(WHR).

Model 3: adjusted for model 2 plus Uric Acid(UA), Triglyceride(TG), Low-Density Lipoprotein Cholesterol(LDL-C) and High-Density Lipoprotein Cholesterol (HDL-C).

P values<0.05 were considered significant.

Based on our analyses, MUAC≥30.9cm was determined as the best cutoff value for men to define population with central obesity, and≥30cm for women, with 92.2% accuracy for men and 78.8% accuracy for women. To our best knowledge, this was the first study to evaluate the association between MUAC and central obesity and to determine the cut-off value of the MUAC for the prediction of central obesity especially in patients with type 2 diabetes.

Another feature of our research was the positive correlation between the MUAC and IR among Chinese type 2 diabetic patients. In our research, we found that MUAC has a positive correlation with HOMA-IR levels, which has been demonstrated to be the indicator of insulin resistance. The relationship between MUAC and IR was also confirmed in recent studies [18,20]. The study in Pakistan [18] focused on children with majority of normal BMI found that arm-fat percentage is positively correlated with insulin levels. In a study conducted among 147 adults with overweight or obesity, Gómez-García et al [20] made a conclusion that mid arm circumference was a better predictor of IR. The subjects in these studies, however, were all not patients with diabetes. Unlike previous studies, we further explored the relationship between MUAC and IR among type 2 diabetes. As diabetes, a metabolic disorder causing disease, has been proved to be a more important factor than obesity in causing IR [36], results in our study further demonstrated that the increase of MUAC could serve as another important indicator of causing IR among diabetic patients. Furthermore, we found MUAC might be superior to WC (Men: $r_{MUAC}$ = 0.52, $r_{WC}$ = 0.44; Women: $r_{MUAC}$ = 0.55, $r_{WC}$ = 0.35) in measuring IR in type 2 diabetes which indicated that MUAC could be used as a better screening method of IR in type 2 diabetes.

We also evaluated the association between MUAC and metabolic risk factors including plasma UA and lipid. We found that MUAC was correlated positively with UA, LDL-C and negatively with HDL-C, indicating a higher risk of developing metabolic disorders in a population with a larger MUAC, which reminded us of taking more concern on the metabolic risk

**Table 6. Logistic regression analysis of risk factors for central obesity.**

|  | Confirmed Central Obesity | | |
|---|---|---|---|
|  | Odds ratio | (95% CI) | P value |
| MUAC (cm) | 2.129 | 1.311, 3.457 | 0.002 |
| LDL-C (mmol/l) | 3.023 | 1.090, 8.383 | 0.033 |

Risk factors including Mid-upper Arm Circumference(MUAC), age, gender, Low-Density Lipoprotein Cholesterol (LDL-C), Triglyceride(TG), Uric Acid(UA) and Use of Statin.

P values<0.05 were considered significant.

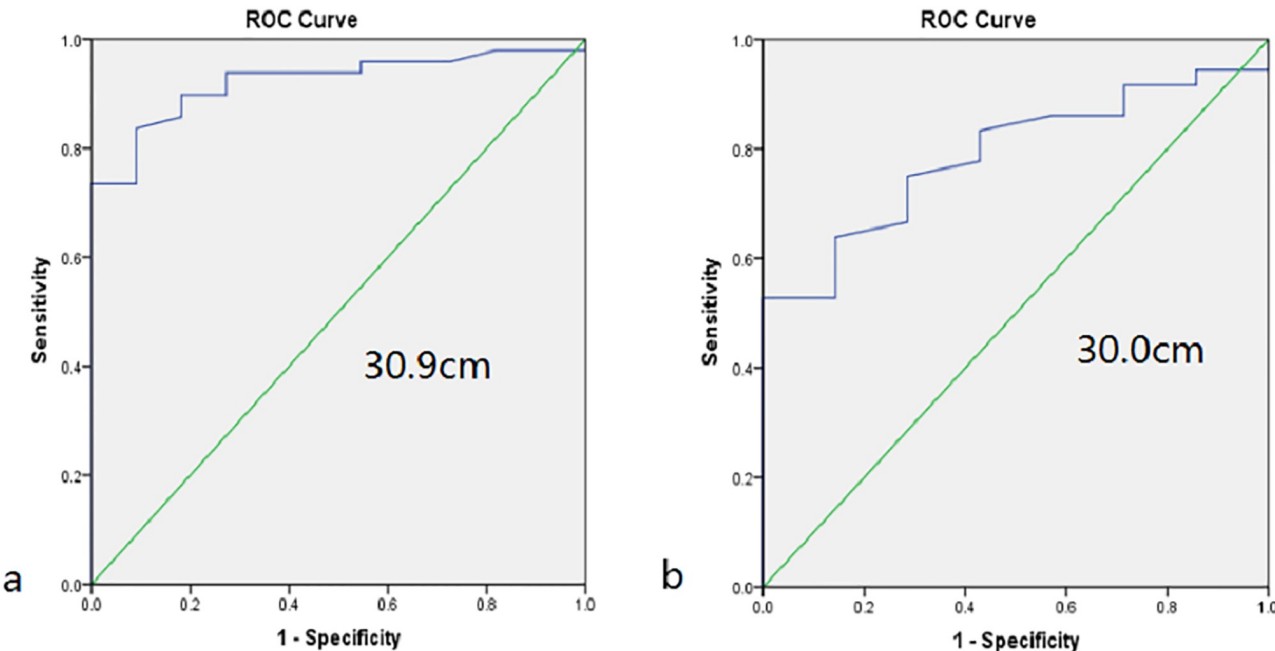

**Fig 2. The receiver operating characteristic (ROC) curves for men and women to identify central obesity. (a)** ROC curve for MUAC in men. AUC = 0.922 ($P < 0.001$), 95% CI 0.852–0.992. Identified MUAC cutoff value = 30.9cm, Youden index = 0.746, Sensitivity: 83.7%; Specificity: 90.9%. **(b)** ROC curve for MUAC in women. AUC = 0.788 ($P = 0.017$), 95% CI 0.642–0.933. Identified MUAC cutoff value = 30.0cm, Youden index = 0.528, Sensitivity: 52.8%; Specificity: 99.9%. MUAC, Mid-upper Arm Circumference. AUC: areas under the curve.

factors in this population. It is interesting that while MUAC correlated with BMI, waist circumference and LDL-C in both genders, but the correlation between MUAC and WHR, Tg, Uric acid, and HDL cholesterol in women was not observed. The plausible reason for sex difference of WHR may be due to the different fat distribution during aging. Men have consistent fat distribution during aging, which is always characterized with more visceral fat in the abdomen (apple shape), but women have more subcutaneous fat in the hip and thighs (pear shape) before menopause and have more visceral fat in abdomen (apple shape) after menopause due to the dramatical decline of estrogen [37]. The female patients in our study, however, were mainly during the menopause period. As a result, no correlation was shown between MUAC and WHR in these estrogen-changing women. The differences of Tg, Uric acid, and HDL cholesterol also attributed to the decline of estrogen after menopause for the protective role of estrogen for metabolic diseases [38]. Therefore, using the MUAC as a clinical tool to detect some metabolic risk factors, such as WHR, Tg, Uric acid, and HDL cholesterol in women should be cautious.

In our study, we found the MUAC was a simple and effective measurement for the determination of central obesity, insulin resistance and multiple metabolic risk factors among type 2 diabetic patients in China. Moreover, we also confirmed the cut-off values of MUAC for central obesity evaluation. Recently, Hou Y, et al also conducted a similar study about MUAC in the Chinese population [39]. However, the study was different from ours. Firstly, the population in their study included both participates with diabetes or normal glucose tolerance (NGT). The analysis of association between MUAC and central obesity and IR was conducted in all the participants but not just especially in patients with diabetes. The only analysis especially involved with diabetes was the multivariable logistic regression analysis base on the

subgroups of diabetes or not. As we mentioned above, the central obesity, IR and MUAC in diabetes were quite different from those in participates with NGT, then the conclusion and significance of Hou's could not be drawn to the population with diabetes. Secondly, the aim of the study was to investigate the associations between MUAC and cardiometabolic risk profiles but not mainly the MUAC and central obesity and IR.

Comparing MUAC with WC and WHR, MUAC has some advantages: without influence of the moment of measurement (no influence of dining); simpler for both physicians and subjects, especially in the public and crowded places; more convenient and more socially acceptable, particularly for obese population; additionally, more accurate of the result for the unified method of detection. As a result, MUAC may be a better index in large epidemiological survey about central obesity and insulin resistance in type 2 diabetes.

Our research has some limitations. Firstly, the sample size in our research was relatively small. However, this is just a pilot study evaluating the association between MUAC and central obesity and IR, especially in patients with type 2 diabetes. Therefore, with the suggestive findings from this study, prospectively designed studies with more participants would be conducted in the near future. Secondly, the index of HOMA-IR, which represents insulin resistance levels, remains relatively simple and no international reference values. However, as we know, HOMA-IR has already been proved to be closely correlated with the hyperinsulinemic euglycemic clamp [40], the gold standard to evaluate IR. Therefore, we think the evaluation of IR in our study might be relatively simple but practical. Thirdly, the anti-diabetic agents and anti-hypertensive agents could influence insulin resistance, which might consequently cause effects on our results. It would be most appropriate to include completely drug-naïve subjects into our study. However, this was a cross-sectional study but not a case-control clinical trial, we could not intervene in any way to stop the anti-diabetic agents or any other medications as clinical trials do. We can also see pharmacotherapy of patients with T2DM, hyperlipidemia or hypertention were not stopped in many similar studies involving the investigation of insulin resistance [41–44]. Moreover, the subjects included in our study were mostly in their middle age. Sixty-eight percent of middle-age and elderly population in our society had at least one chronic disease [45]. In addition, the prevalence of hypertension and hyperlipidemia in patients with type 2 diabetes is 51.9% and 30.5% respectively [46–47]. As a society with dramatically increased aging population, we can hardly include the subjects at the middle age without any other diseases or any other drugs. Drug-naïve subjects will be included into our future study to minimize the effects of agents on insulin resistance. Despite the limitations, our results offered evidence that MUAC might be a screening tool that could predict central obesity and IR, which might be superior to WC and WHR in Chinese patients with type 2 diabetes.

In conclusion, higher MUAC is correlated with central obesity, insulin resistance and multiple metabolic risk indicators in Chinese subjects with type 2 diabetes. It could be expectable that MUAC will be widely applied in both research and clinical practice in future for its simplicity and stability.

## Supporting information

**S1 Table. Hypoglycemic agents.**
(DOCX)

**S2 Table. Hypotensive agents.**
(DOCX)

**S3 Table. Lipid-lowering agents.**
(DOCX)

**S4 Table. Uric-acid-lowering agents.**
(DOCX)

## Acknowledgments

The authors wish to thank the researchers for their assistance of measurement and samples collection. We also thank all the members of our team for their contribution and the subjects who participated in our study.

Bin Yao and Xu-bin Yang developed the idea and designed the research. Yanhua Zhu and Qiong-yan Lin collected and analyzed the data, wrote the manuscript. Yao Zhang analyzed the data and revised the manuscript. Hong-rong Deng collected the data and contributed to the discussion. Xi-ling Hu helped to analyze the data.

## Author Contributions

**Data curation:** Yanhua Zhu, Qiongyan Lin, Hongrong Deng.

**Formal analysis:** Yanhua Zhu, Yao Zhang, Hongrong Deng, Xiling Hu.

**Funding acquisition:** Bin Yao.

**Investigation:** Qiongyan Lin.

**Project administration:** Xubin Yang, Bin Yao.

**Resources:** Xubin Yang, Bin Yao.

**Writing – original draft:** Qiongyan Lin.

**Writing – review & editing:** Yao Zhang, Xubin Yang.

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
