## [Decision Letter · Decision Letter 0]

14 Oct 2019

PONE-D-19-19096

Mid-upper Arm Circumference as a Simple Tool for Identifying Central Obesity and Insulin Resistance in Type 2 Diabetes

PLOS ONE

Dear Dr Yao,

Thank you for submitting your manuscript to PLOS ONE. After careful consideration, we feel that it has merit but does not fully meet PLOS ONE’s publication criteria as it currently stands. Therefore, we invite you to submit a revised version of the manuscript that addresses the points raised during the review process.

We would appreciate receiving your revised manuscript by Nov 28 2019 11:59PM. To enhance the reproducibility of your results, we recommend that if applicable you deposit your laboratory protocols in protocols.io, where a protocol can be assigned its own identifier (DOI) such that it can be cited independently in the future. For instructions see: http://journals.plos.org/plosone/s/submission-guidelines#loc-laboratory-protocols

We look forward to receiving your revised manuscript.

Kind regards,

Mauro Lombardo

Academic Editor

PLOS ONE

Journal Requirements:

2. Please include your ethics statement from your methods section in the online metadata.

Additional Editor Comments:

Dear authors. Thank you for your work submitted to the journal. Please reply point by point to the comments of the reviewers. Regards

Reviewers' comments:

Reviewer's Responses to Questions

**Comments to the Author**

1. Is the manuscript technically sound, and do the data support the conclusions?

Reviewer #1: Partly

Reviewer #2: Partly

2. Has the statistical analysis been performed appropriately and rigorously? 

Reviewer #1: I Don't Know

Reviewer #2: Yes

3. Have the authors made all data underlying the findings in their manuscript fully available?

Reviewer #1: No

Reviewer #2: No

4. Is the manuscript presented in an intelligible fashion and written in standard English?

Reviewer #1: No

Reviewer #2: Yes

5. Review Comments to the Author

Reviewer #1: This paper investigated the utility of measuring mid-upper arm circumference (MUAC) as a surrogate for measuring waist circumference or waist to hips ratio when assessing insulin resistance in patients with type 2 diabetes. It is concluded that MUAC is a simple tool to measure central obesity and insulin resistance. It is of interest that while MUAC correlated with BMI, waist circumference and and LDL-C in both genders, It failed to correlate with WHR, Tg, Uric acid, and HDL cholesterol in women. The implication of this failure in women was not discussed adequately. The possible reason (women are pear or apple shape while men are always apple shape) and possible consequence of using the MUAC as a clinical tool in women was not discussed.

On Page 19 Line 3 it is stated that "Furthermore we found MUAC might be superior to WC in measuring IR in type 2 diabetes..." It would be helpful to the reader to quote here the data and statistical result that support this statement.

Minor Points

Page 4 Para 2 Line 3 "...it remains a number of limitations od WC.." should be "....there remain a number of limitations of WC.."

Page 4 Para 2 Line 11 "Therefore some Researches were conducted....." should be "Therefore som research was conducted..."

Page 4 Para 2 Line 15 "However it remains little data to..." should be "However there is little data to..."

There are many other examples where a native English speaker might help.

Reviewer #2: Dear authors,

This article is interesting because authors investigated the importance MUAC. However I think this article has unavoidable problems to be confirmed.

<1> Considering this study was performed for T2DM patients, authors should investigate the effect of T2DM itself to results in detail. Authors wrote “without insulin treatment or other medications that could alter insulin secretion (such as Sulphonylurea) for at least 2 weeks, not having severe disease and be free of any acute infection during 2 weeks before the inclusion” at MATERIALS and METHODS and patients’ HbA1c levels in Table 1. At first, authors should write details of agents for T2DM which patients were administered when they were investigated. For instance, almost all agents for T2DM affected to insulin profile, DPP4Is and/or SGLT2Is particularly (of course, glinide were not used, I think). If patients used these two agents in particular, it must affect the results (If patients did not use these two agents, authors should write detail of agents for T2DM in Table 1). Indeed, authors should investigate the association between MUAC and glycemic parameters because authors targeted T2DM patients. If authors are thinking there is no necessity to consider glycemic parameters in this study, this study’s aim could not be understood by many of readers (including me) as they will not understand the difference between this study and the previous article. Moreover, authors must see the article, [Hou Y, et al. BMJ Open 2019; 9: e028904. doi:10.1136/bmjopen-2019-028904]. This article included 6287 participants with or without diabetes aged 40 years or older and was investigated from very various viewpoints. This article has already revealed almost all of this study’s results except the participants for not only T2DM. Actually, I think this article did not have enough novelty, and this study cohort was not large. So what I wrote above must be investigated and mentioned in order to gain novelty; this study focused only T2DM patients.

In addition, medications of HT can also affect the results. Moreover, if patients were medicated by agents against hyperlipidemia and hyperuricemia, these must affect the results definitely. Authors should also investigate and mention them.

<2> Authors should explain log HOMAIR were used to investigation of association with MUAC. I would like to know why authors did not use HOMAIR itself. I would like to know whether authors considered statistical problem with using HOMAIR itself or not.

In summary, this article surely targeted T2DM patients, but more novelty were required. Statistical analysis itself was done properly, but considering novelty of this article or descriptions, it is insufficient, I think. Almost all of this article were very similar to the previous reports of non T2DM patients. Readers might think targeting only T2DM patients did not make sense at investigations like this. So, authors should reveal the impotence of this study properly. If authors could not confirm above all, this article is difficult to be accepted.

Regards,

6. PLOS authors have the option to publish the peer review history of their article (what does this mean?). If published, this will include your full peer review and any attached files.

Reviewer #1: No

Reviewer #2: No

---

## [Author Response · Author response to Decision Letter 0]

31 Dec 2019

Reviewer #2: Dear authors,

This article is interesting because authors investigated the importance MUAC. However I think this article has unavoidable problems to be confirmed.

1、 Considering this study was performed for T2DM patients, authors should investigate the effect of T2DM itself to results in detail. Authors wrote “without insulin treatment or other medications that could alter insulin secretion (such as Sulphonylurea) for at least 2 weeks, not having severe disease and be free of any acute infection during 2 weeks before the inclusion” at MATERIALS and METHODS and patients’ HbA1c levels in Table 1. At first, authors should write details of agents for T2DM which patients were administered when they were investigated. For instance, almost all agents for T2DM affected to insulin profile, DPP4Is and/or SGLT2Is particularly (of course, glinide were not used, I think). If patients used these two agents in particular, it must affect the results (If patients did not use these two agents, authors should write detail of agents for T2DM in Table 1). 

Answer:

Thank you for your comment.

We’ve added the details of hypoglycemic agents in the MATERIALS AND METHODS and supplemental table 1 as you suggested. All the patients were treated without insulin treatment, Sulphonylurea, sodium-dependent glucose transporters-2 inhibitors (SGLT-2i) or glucagon-like peptide-1 (GLP-1) analog for at least 2 weeks. (see Paragraph 2 in Page 5 of the resubmitted manuscript with tracked changes)

2、Indeed, authors should investigate the association between MUAC and glycemic parameters because authors targeted T2DM patients. If authors are thinking there is no necessity to consider glycemic parameters in this study, this study’s aim could not be understood by many of readers (including me) as they will not understand the difference between this study and the previous article. Moreover, authors must see the article, [Hou Y, et al. BMJ Open 2019; 9: e028904. doi:10.1136/bmjopen-2019-028904]. This article included 6287 participants with or without diabetes aged 40 years or older and was investigated from very various viewpoints. This article has already revealed almost all of this study’s results except the participants for not only T2DM. Actually, I think this article did not have enough novelty, and this study cohort was not large. So what I wrote above must be investigated and mentioned in order to gain novelty; this study focused only T2DM patients.

Answer:

Thank you for your great comment.

We’ve added the analysis of the association between MUAC and glycemic parameters in the RESULTS, table 1, table 2 and table3 (see Paragraph 2 in Page 8 of the resubmitted manuscript with tracked changes). However, the association between MUAC and glycemic parameters was not observed in our research.

We acknowledged that Hou’s study also focused on MUAC in the Chinese population. However, the study was different from ours. Firstly, the population in their study included both participates with diabetes or normal glucose tolerance (NGT). The analysis of association between MUAC and central obesity or IR was conducted in all the participants but not just in patients with diabetes. The only analysis specially involved with diabetes was the multivariable logistic regression analysis base on the subgroups of diabetes or not. The key points that two studies concerned were different. As the importance of IR in diabetes was reinforced in the most recent studies1-2 and the central obesity, IR and MUAC in diabetes were quite different from those in participants with NGTs3, the conclusion and significance of Hou’s study could not be drawn to the population with diabetes. Secondly, the aim of Hou’s study was to investigate the associations between MUAC and cardiometabolic risk profiles but not mainly the MUAC and central obesity and IR. 

The relevant contents were added in the INTRODUCTION (see Paragraph 1 in Page 4 of the resubmitted manuscript with tracked changes) and DISCUSSION (see Paragraph 2 in Page 17 of the resubmitted manuscript with tracked changes).

We acknowledged that the sample size in our research was relatively small. However, this is just a pilot study evaluating the association between MUAC and central obesity and IR specially in patients with diabetes. With the suggestive findings from this study, prospectively designed studies with more participants would be conducted in the near future. The relevant contents were added in the DISCUSSION (see Paragraph 3 in Page 18 of the resubmitted manuscript with tracked changes).

[1] AHLQVIST E, STORM P, KARAJAMAKI A, et al. 2018. Novel subgroups of adult-onset diabetes and their association with outcomes: a data-driven cluster analysis of six variables[J]. Lancet Diabetes Endocrinol, 6(5): 361-369.

[2] ZOU X, ZHOU X, ZHU Z, et al. 2019. Novel subgroups of patients with adult-onset diabetes in Chinese and US populations[J]. Lancet Diabetes Endocrinol, 7(1): 9-11.

[3] YANG Q, GRAHAM T E, MODY N, et al. 2005. Serum retinol binding protein 4 contributes to insulin resistance in obesity and type 2 diabetes[J]. Nature, 436(7049): 356-362.

3、In addition, medications of HT can also affect the results. Moreover, if patients were medicated by agents against hyperlipidemia and hyperuricemia, these must affect the results definitely. Authors should also investigate and mention them.

Answer:

Thank you for your comment.

We have added the details of the medications of hypertension, hyperlipidemia and hyperuricemia in the supplemental table 2, table3 and table4, respectively.

4、Authors should explain log HOMAIR were used to investigation of association with MUAC. I would like to know why authors did not use HOMAIR itself. I would like to know whether authors considered statistical problem with using HOMAIR itself or not.

Answer:

Thank you for your comment.

As we mentioned in the Statistical analysis (see Paragraph 1 in Page 7 of the resubmitted manuscript with tracked changes), data of non-normal distributions were logarithmically transformed before statistical analysis. 

As a result, to ensure normality of distribution and to meet the criteria for regression analysis, we used log HOMAIR to investigate the association of HOMAIR and MUAC as many other studies did1-3.

[1] SANTHANAM P, ROWE S P, DIAS J P, et al. 2019. Relationship between DXA measured metrics of adiposity and glucose homeostasis; An analysis of the NHANES data[J]. PLoS One, 14(5): e0216900..

[2] MENTE A, MEYRE D, LANKTREE M B, et al. 2013. Causal relationship between adiponectin and metabolic traits: a Mendelian randomization study in a multiethnic population[J]. PLoS One, 8(6): e66808.

[3] ISHIMURA S, FURUHASHI M, WATANABE Y, et al. 2013. Circulating levels of fatty acid-binding protein family and metabolic phenotype in the general population[J]. PLoS One, 8(11): e81318.

Reviewer #1: 

1、This paper investigated the utility of measuring mid-upper arm circumference (MUAC) as a surrogate for measuring waist circumference or waist to hips ratio when assessing insulin resistance in patients with type 2 diabetes. It is concluded that MUAC is a simple tool to measure central obesity and insulin resistance. It is of interest that while MUAC correlated with BMI, waist circumference and and LDL-C in both genders, It failed to correlate with WHR, Tg, Uric acid, and HDL cholesterol in women. The implication of this failure in women was not discussed adequately. The possible reason (women are pear or apple shape while men are always apple shape) and possible consequence of using the MUAC as a clinical tool in women was not discussed.

Answer:

Thank you for your excellent comment.

It is interesting that while MUAC correlated with BMI, waist circumference and and LDL-C in both genders, but the correlation betwee MUAC and WHR, Tg, Uric acid, and HDL cholesterol in women was not observed. The plausible reason for this sex difference may be due to the different fat distribution during aging. Men have consistent fat distribution during aging, which is always characterized with more visceral fat in the abdomen (apple shape), but women have more subcutaneous fat in the hip and thighs(pear shape) before menopause and have more visceral fat in abdomen (apple shape) after menopause due to the dramatical decline of estrogen1. The female patients in our study, however, were mainly during the menopause period. As a result, no correlation was shown between MUAC and WHR in these estrogen-changing women. The differences of Tg, Uric acid, and HDL cholesterol also attributed to the decline of estrogen after menopause for the protective role of estrogen for metabolic diseases2. Therefore, using the MUAC as a clinical tool to detect some metabolic risk factors, such as WHR, Tg, Uric acid, and HDL cholesterol in women should be cautious. 

 The relevant contents were added in the DISCUSSION (see Paragraph 2 in Page 16 and Paragraph 1 in Page 17 of the resubmitted manuscript with tracked changes).

[1] KARASTERGIOU K, SMITH S R, GREENBERG A S, et al. 2012. Sex differences in human adipose tissues - the biology of pear shape[J]. Biol Sex Differ, 3(1): 13.

[2] TRAMUNT B, SMATI S, GRANDGEORGE N, et al. 2019. Sex differences in metabolic regulation and diabetes susceptibility[J]. Diabetologia.

2、On Page 19 Line 3 it is stated that "Furthermore we found MUAC might be superior to WC in measuring IR in type 2 diabetes..." It would be helpful to the reader to quote here the data and statistical result that support this statement.

Answer:

Thank you for your comment.

We’ve added the data and statistical result on 16 Page Line 1. 

3、Minor Points

Page 4 Para 2 Line 3 "...it remains a number of limitations od WC.." should be "....there remain a number of limitations of WC.."

Page 4 Para 2 Line 11 "Therefore some Researches were conducted....." should be "Therefore som research was conducted..."

Page 4 Para 2 Line 15 "However it remains little data to..." should be "However there is little data to..."

There are many other examples where a native English speaker might help.

 Answer:

We are sincerely sorry about the language issue. We have revised some mistakes as you suggested. 

We’ll seek help from English editing company and thoroughly review and revise the manuscript in the next revision.

---

## [Decision Letter · Decision Letter 1]

9 Jan 2020

PONE-D-19-19096R1

Mid-upper Arm Circumference as a Simple Tool for Identifying Central Obesity and Insulin Resistance in Type 2 Diabetes

PLOS ONE

Dear Dr Yao,

Thank you for submitting your manuscript to PLOS ONE. After careful consideration, we feel that it has merit but does not fully meet PLOS ONE’s publication criteria as it currently stands. Therefore, we invite you to submit a revised version of the manuscript that addresses the points raised during the review process.

We would appreciate receiving your revised manuscript by Feb 23 2020 11:59PM. To enhance the reproducibility of your results, we recommend that if applicable you deposit your laboratory protocols in protocols.io, where a protocol can be assigned its own identifier (DOI) such that it can be cited independently in the future. For instructions see: http://journals.plos.org/plosone/s/submission-guidelines#loc-laboratory-protocols

We look forward to receiving your revised manuscript.

Kind regards,

Mauro Lombardo

Academic Editor

PLOS ONE

Additional Editor Comments (if provided):

Please answer to the 1st reviewer's comments and resubmit

Reviewers' comments:

Reviewer's Responses to Questions

**Comments to the Author**

1. If the authors have adequately addressed your comments raised in a previous round of review and you feel that this manuscript is now acceptable for publication, you may indicate that here to bypass the “Comments to the Author” section, enter your conflict of interest statement in the “Confidential to Editor” section, and submit your "Accept" recommendation.

Reviewer #1: All comments have been addressed

Reviewer #2: (No Response)

2. Is the manuscript technically sound, and do the data support the conclusions?

Reviewer #1: Yes

Reviewer #2: Partly

3. Has the statistical analysis been performed appropriately and rigorously? 

Reviewer #1: I Don't Know

Reviewer #2: No

4. Have the authors made all data underlying the findings in their manuscript fully available?

Reviewer #1: Yes

Reviewer #2: No

5. Is the manuscript presented in an intelligible fashion and written in standard English?

Reviewer #1: Yes

Reviewer #2: Yes

6. Review Comments to the Author

Reviewer #1: Thank you for addressing my concerns. Therefore it is now suitable for publication. I don't know if raw data has been made available.

Reviewer #2: Dear authors,

I think authors reviewed and answered against my comment carefully. Meanwhile, there are some remaining important problems, especially in the data authors added.

I commented to authors about the novelty of this study considering the previous article with large numbers. Actually I understand there is the difference between this and the previous article with regard to the subjects (the subjects of this study were only patients of T2DM). However, partly because this study cohort were not large, I commented to authors about various points.

At first, authors should consider the effect to results by the administration of subjects more. The supplemental table which authors added showed the difficulty of proving correctness of authors’ investigation and conclusion. I wrote that almost all agents for T2DM affected to HOMA-IR, especially not only sulphonylurea/glinide but also DPP4i /SGLT2i affected to insulin profile itself previously. Actually, authors could not remove the possibility of effect of anti-diabetic agents to the results and analysis of HOMA-IR. I even think it is possible diuretics or ARB/ACEi administration affected HOMA-IR.

Considering authors’ opinion of the novelty of this study (the subjects of this study were only patients of T2DM), authors must remove the possibility of effect of anti-diabetic agents at least.

Same as above, authors should reconsider table 6 and related results. LDL-C must be affected by statin. 70 of 103 patients were with statin in this study. Authors must consider the effect of statin to the results and analysis.

Honestly, I think all subjects should be without anti-diabetic agents. Furthermore, if authors would like to mention about the relationships including LDL-C, the subjects should be without agents against hyperlipidemia or authors should investigate and analyze considering the effect of agents against hyperlipidemia.

I would like authors to understand my previous and present comments adequately.

Authors’ revision considering these important points must be needed for this article to be accepted.

Regards,

7. PLOS authors have the option to publish the peer review history of their article (what does this mean?). If published, this will include your full peer review and any attached files.

Reviewer #1: No

Reviewer #2: No

---

## [Author Response · Author response to Decision Letter 1]

20 Feb 2020

Point-by –Point Answers to 

the Editor’s and Reviewers’ comments

Reviewer #2: Dear authors,

I think authors reviewed and answered against my comment carefully. Meanwhile, there are some remaining important problems, especially in the data authors added.

I commented to authors about the novelty of this study considering the previous article with large numbers. Actually I understand there is the difference between this and the previous article with regard to the subjects (the subjects of this study were only patients of T2DM). However, partly because this study cohort were not large, I commented to authors about various points.

At first, authors should consider the effect to results by the administration of subjects more. The supplemental table which authors added showed the difficulty of proving correctness of authors’ investigation and conclusion. I wrote that almost all agents for T2DM affected to HOMA-IR, especially not only sulphonylurea/glinide but also DPP4i /SGLT2i affected to insulin profile itself previously. Actually, authors could not remove the possibility of effect of anti-diabetic agents to the results and analysis of HOMA-IR. I even think it is possible diuretics or ARB/ACEi administration affected HOMA-IR.

Considering authors’ opinion of the novelty of this study (the subjects of this study were only patients of T2DM), authors must remove the possibility of effect of anti-diabetic agents at least.

Same as above, authors should reconsider table 6 and related results. LDL-C must be affected by statin. 70 of 103 patients were with statin in this study. Authors must consider the effect of statin to the results and analysis.

Honestly, I think all subjects should be without anti-diabetic agents. Furthermore, if authors would like to mention about the relationships including LDL-C, the subjects should be without agents against hyperlipidemia or authors should investigate and analyze considering the effect of agents against hyperlipidemia.

I would like authors to understand my previous and present comments adequately.

Authors’ revision considering these important points must be needed for this article to be accepted.

Regards,

Answer:

Thank you so much for your excellent comments.

First，honestly, we were very clear that all the pharmacotherapy of T2DM, hyperlipidemia or hypertention could definitely influenced insulin resistance. However, this was a cross-sectional study but not a case-control clinical trial, we could not intervene in any way to stop the anti-diabetic agents or any other medications as clinical trials do [1-2]. We can also see pharmacotherapy of patients with T2DM, hyperlipidemia or hypertention were not stopped in many similar studies involving the investigation of insulin resistance [3-5].

Second, we should admit that we could not completely eliminate all the influencing factors. For example, if we stopped the pharmacotherapy before evaluating HOMA-IR (note that the mean diabetes duration of patients in our study was 8~9 years and the mean HbA1c was 9%), the plasma glucose would elevate, sometimes to a very high level. As is well known, the strongest insulin secretion influencing factor is glucose [6]. As a result, the elevating glucose would definitely influence the levels of insulin, and consequently influence HOMA-IR. Therefore, stopping the pharmacotherapy will still influence HOMA-IR. 

Besides, in the guidelines of American Diabetes Association (ADA), European Association for the Study of Diabetes (EASD) or Chinese Diabetes Society (CDS), lifestyle intervention was recommended throughout the management of T2DM as a fundamental treatment [7-8]. The lifestyle management can also significantly change the insulin resistance [9]. However, subjects in studies with pharmacotherapy or not could not stop lifestyle management, which consequently could not avoid its effect on HOMA-IR.

That may be the reason why the real-world study, with more diverse settings but less intervention, is gaining more and more concern [10].

Third, as you pointed out that it is possible diuretics or ARB/ACEi administration could affect HOMA-IR, it would be most appropriate to include completely drug-naïve subjects into our study. However, the subjects included in our study were mostly in their middle age. Sixty-eight percent of middle-age and elderly population in our society had at least one chronic disease[11]. In addition, the prevalence of hypertension and hyperlipidemia in patients with diabetes is 51.9% and 30.5% respectively. More than 50% diabetic patients had at least one chronic diabetes complication [12-13]. As a society with dramatically increased aging population, we can hardly include the subjects at the middle age without any other diseases or any other drugs.

For the reasons above, we can not eliminate all the affecting factors including the effects of drugs or lifestyle intervention whether the cohort is large or not. As we mentioned in the discussion “ this is just a pilot study evaluating the association between MUAC and central obesity and IR, specially in patients with diabetes. Therefore, with the suggestive findings from this study, prospectively designed studies with more participants would be conducted in the near future” (see Paragraph 3 in Page 18 of the resubmitted manuscript with tracked changes), prospectively designed studies with drug-naive participants would be conducted in the near future at your great suggestions.

For the question of LDL-C affected by statin, besides the reasons mentioned above, we also conducted a logistic regression to adjust for using statin or not and find that using statin or not did not affect the results (OR：3.023 and P：0.033). (see Paragraph 3 in Page 11 of the resubmitted manuscript with tracked changes)

[1] SEDGWICK P. 2015. Bias in observational study designs: cross sectional studies[J]. Bmj, 350: h1286.

[2] Kesmodel, Ulrik, S. Cross‐sectional studies – what are they good for?[J]. Acta Obstetricia et Gynecologica Scandinavica: Official Publication of the Nordisk Forening for Obstetrik och Gynekologi, 2018.

[3] Mamtani M , Kulkarni H , Dyer T D , et al. Waist Circumference Independently Associates with the Risk of Insulin Resistance and Type 2 Diabetes in Mexican American Families[J]. PLOS ONE, 2013, 8.

[4] Lim J S , Choi Y J , Kim S K , et al. Optimal Waist Circumference Cutoff Value Based on Insulin Resistance and Visceral Obesity in Koreans with Type 2 Diabetes[J]. Diabetes & Metabolism Journal, 2015, 39(3).

[5] Elkeles R S , Godsland I F , Feher M D , et al. Coronary calcium measurement improves prediction of cardiovascular events in asymptomatic patients with type 2 diabetes: the PREDICT study[J]. European Heart Journal, 2008, 29(18):2244-2251.

[6]Jackson R , Rudelt C , Willaime J P . Effects of prolonged glucose infusion on insulin secretion, clearance, and action in normal subjects.[J]. American Journal of Physiology, 1996, 270(2 Pt 1):E251

[7]2020. 6. Glycemic Targets: Standards of Medical Care in Diabetes-2020[J]. Diabetes Care, 43(Suppl 1): S66-s76.

[8]Bailey, Timothy. Options for Combination Therapy in Type 2 Diabetes: Comparison of the ADA/EASD Position Statement and AACE/ACE Algorithm[J]. The American Journal of Medicine, 2013, 126(9):S10-S20.

[9]SAMPATH KUMAR A, MAIYA A G, SHASTRY B A, et al. 2019. Exercise and insulin resistance in type 2 diabetes mellitus: A systematic review and meta-analysis[J]. Ann Phys Rehabil Med, 62(2): 98-103.

[10] Sherman R E , Anderson S A , Dal Pan G J , et al. Real-World Evidence — What Is It and What Can It Tell Us?[J]. New England Journal of Medicine, 2016, 375(23):2293-2297.

[11] Cheng Y，Cao Z，Hou J , et al. Investigation and association analysis of multimorbidity in middle-aged and elderly population in China, 2019, 23( 6):625-629.

[12] Ji L , Hu D , Pan C , et al. Primacy of the 3B Approach to Control Risk Factors for Cardiovascular Disease in Type 2 Diabetes Patients[J]. The American Journal of Medicine, 2013, 126(10):925.e11-925.e22. 

[13] Liu Z , Fu C , Wang W , et al. Prevalence of chronic complications of type 2 diabetes mellitus in outpatients - a cross-sectional hospital based survey in urban China[J]. Health & Quality of Life Outcomes, 2010, 8(1):62-0. 

Reviewer #1: Thank you for addressing my concerns. Therefore it is now suitable for publication. I don't know if raw data has been made available.

Answer:

Thank you for your comment.

Raw data has been available now.

---

## [Decision Letter · Decision Letter 2]

27 Feb 2020

PONE-D-19-19096R2

Mid-upper Arm Circumference as a Simple Tool for Identifying Central Obesity and Insulin Resistance in Type 2 Diabetes

PLOS ONE

Dear Dr Yao,

Thank you for submitting your manuscript to PLOS ONE. After careful consideration, we feel that it has merit but does not fully meet PLOS ONE’s publication criteria as it currently stands. Therefore, we invite you to submit a revised version of the manuscript that addresses the points raised during the review process.

We would appreciate receiving your revised manuscript by Apr 12 2020 11:59PM. To enhance the reproducibility of your results, we recommend that if applicable you deposit your laboratory protocols in protocols.io, where a protocol can be assigned its own identifier (DOI) such that it can be cited independently in the future. For instructions see: http://journals.plos.org/plosone/s/submission-guidelines#loc-laboratory-protocols

We look forward to receiving your revised manuscript.

Kind regards,

Mauro Lombardo

Academic Editor

PLOS ONE

Reviewers' comments:

Reviewer's Responses to Questions

**Comments to the Author**

1. If the authors have adequately addressed your comments raised in a previous round of review and you feel that this manuscript is now acceptable for publication, you may indicate that here to bypass the “Comments to the Author” section, enter your conflict of interest statement in the “Confidential to Editor” section, and submit your "Accept" recommendation.

Reviewer #1: All comments have been addressed

Reviewer #2: (No Response)

2. Is the manuscript technically sound, and do the data support the conclusions?

Reviewer #1: Yes

Reviewer #2: Yes

3. Has the statistical analysis been performed appropriately and rigorously? 

Reviewer #1: Yes

Reviewer #2: Yes

4. Have the authors made all data underlying the findings in their manuscript fully available?

Reviewer #1: (No Response)

Reviewer #2: Yes

5. Is the manuscript presented in an intelligible fashion and written in standard English?

Reviewer #1: Yes

Reviewer #2: Yes

6. Review Comments to the Author

Reviewer #1: (No Response)

Reviewer #2: Dear authors,

I am glad to see authors careful and proper answers to my comment.　

Firstly, authors performed modified logistic regression (adjusting for using statin or not) adequately. This must be needed and I am really glad to see this result.

In addition, authors wrote “this is just a pilot study evaluating the association between MUAC and central obesity and IR, specially in patients with diabetes” in discussion. I have been thinking authors should mention this, and authors did in this revised document. I am also glad to see the description.

Regarding effect of antidiabetic agents to results, I understand authors’ reply. As authors wrote in answers to my comments, this study was cross-sectional study. I think the study cohort was small as cross-sectional study. Of course, I understood anti-diabetic agents (as well as anti-hypertensive agents) should not be stopped in patients of T2DM. On the other hand, the effect to results should be considered because of this small cohort especially.

Considering above, authors should mention the limitation of this study more specifically. In detail, authors should add the description about the possibility of effect of anti-diabetic agents and anti-hypertensive agents to results, same as authors wrote in answers to my comments. Limitations should be written adequately in order to be understood by readers properly.

The revision of this article must be meaningful for authors as well as readers of this article. I really think authors considered my comments devotedly and modified properly. This must be the last recommendation to authors. I am looking forward to seeing authors’ revised document considering my comments.

[additional]

The description which I mentioned above, “this is just a pilot study evaluating the association between MUAC and central obesity and IR, specially in patients with diabetes”, should be modified to “this is just a pilot study evaluating the association between MUAC and central obesity and IR, “especially” in patients with “type 2 diabetes””. Same as this, the other descriptions, “patients with diabetes”, should be modified to “patients with type 2 diabetes”.

Sincerely,

7. PLOS authors have the option to publish the peer review history of their article (what does this mean?). If published, this will include your full peer review and any attached files.

Reviewer #1: No

Reviewer #2: No

---

## [Author Response · Author response to Decision Letter 2]

6 Mar 2020

Thank you for your comments.

1.We have added the limitation of the effects of anti-diabetic agents and anti-hypertensive agents on our results in Discussion.(see Paragraph 1 in Page 19 of the resubmitted manuscript with tracked changes)

2.We have revised some mistakes as you suggested in [additional].

Regards.

---

## [Decision Letter · Decision Letter 3]

23 Mar 2020

Mid-upper Arm Circumference as a Simple Tool for Identifying Central Obesity and Insulin Resistance in Type 2 Diabetes

PONE-D-19-19096R3

Dear Dr. Yao,

We are pleased to inform you that your manuscript has been judged scientifically suitable for publication and will be formally accepted for publication once it complies with all outstanding technical requirements.

With kind regards,

Mauro Lombardo

Academic Editor

PLOS ONE

Additional Editor Comments (optional):

Reviewers' comments:

Reviewer's Responses to Questions

**Comments to the Author**

1. If the authors have adequately addressed your comments raised in a previous round of review and you feel that this manuscript is now acceptable for publication, you may indicate that here to bypass the “Comments to the Author” section, enter your conflict of interest statement in the “Confidential to Editor” section, and submit your "Accept" recommendation.

Reviewer #2: All comments have been addressed

2. Is the manuscript technically sound, and do the data support the conclusions?

Reviewer #2: Yes

3. Has the statistical analysis been performed appropriately and rigorously? 

Reviewer #2: Yes

4. Have the authors made all data underlying the findings in their manuscript fully available?

Reviewer #2: Yes

5. Is the manuscript presented in an intelligible fashion and written in standard English?

Reviewer #2: Yes

6. Review Comments to the Author

Reviewer #2: Dear authors,

I am very glad to see your modified article. Your subimission should be accepted.

I would like you to perform this investigation with large cohort or patients at least not medicated by anti-diabeteic agents in future. Based on this pilot study, I hope this new available method is known widely and finally become routine tests in patients with type 2 diabetes by the future study from you and co-workers.

Sincerely,

7. PLOS authors have the option to publish the peer review history of their article (what does this mean?). If published, this will include your full peer review and any attached files.

Reviewer #2: No

---

## [Editor Report · Acceptance letter]

4 May 2020

PONE-D-19-19096R3 

Mid-upper Arm Circumference as a Simple Tool for Identifying Central Obesity and Insulin Resistance in Type 2 Diabetes 

Dear Dr. Yao:

I am pleased to inform you that your manuscript has been deemed suitable for publication in PLOS ONE. Congratulations! Your manuscript is now with our production department. 

With kind regards,

on behalf of

Dr. Mauro Lombardo 

Academic Editor

PLOS ONE